# Machine Learning Confirms Nonlinear Relationship between Severity of Peripheral Arterial Disease, Functional Limitation and Symptom Severity

**DOI:** 10.3390/diagnostics10080515

**Published:** 2020-07-24

**Authors:** Zulfiqar Qutrio Baloch, Syed Ali Raza, Rahul Pathak, Luke Marone, Abbas Ali

**Affiliations:** 1Department of Cardiology, Michigan State University/Sparrow Hospital, 1215 E Michigan Ave, Lansing, MI 48912, USA; 2Department of Neurology, Emory University, Atlanta, GA 30322, USA; alisyedraza1@gmail.com; 3Heart of Florida Regional Medical Center, UCF College of Medicine, Orlando, FL 32827, USA; rahul.pathak@ucf.edu; 4Heart and Vascular Institute, West Virginia University, Morgantown, WV 26506, USA; luke.marone@wvumedicine.org (L.M.); abbas.ali@hsc.wvu.edu (A.A.)

**Keywords:** Machine Learning, TBI (toe–brachial index), PAD (peripheral arterial disease), CLI (critical limb ischemia), 6MWT (six-minute walk test)

## Abstract

Background: Peripheral arterial disease (PAD) involves arterial blockages in the body, except those serving the heart and brain. We explore the relationship of functional limitation and PAD symptoms obtained from a quality-of-life questionnaire about the severity of the disease. We used a supervised artificial intelligence-based method of data analyses known as machine learning (ML) to demonstrate a nonlinear relationship between symptoms and functional limitation amongst patients with and without PAD. Objectives: This paper will demonstrate the use of machine learning to explore the relationship between functional limitation and symptom severity to PAD severity. Methods: We performed supervised machine learning and graphical analysis, analyzing 703 patients from an administrative database with data comprising the toe–brachial index (TBI), baseline demographics and symptom score(s) derived from a modified vascular quality-of-life questionnaire, calf circumference in centimeters and a six-minute walk (distance in meters). Results: Graphical analysis upon categorizing patients into critical limb ischemia (CLI), severe PAD, moderate PAD and no PAD demonstrated a decrease in walking distance as symptoms worsened and the relationship appeared nonlinear. A supervised ML ensemble (random forest, neural network, generalized linear model) found symptom score, calf circumference (cm), age in years, and six-minute walk (distance in meters) to be important variables to predict PAD. Graphical analysis of a six-minute walk distance against each of the other variables categorized by PAD status showed nonlinear relationships. For low symptom scores, a six-minute walk test (6MWT) demonstrated high specificity for PAD. Conclusions: PAD patients with the greatest functional limitation may sometimes be asymptomatic. Patients without PAD show no relationship between functional limitation and symptoms. Machine learning allows exploration of nonlinear relationships. A simple linear model alone would have overlooked or considered such a nonlinear relationship unimportant.

## 1. Introduction

Peripheral arterial disease (PAD) is a chronic atherosclerotic disease that obstructs the peripheral arteries mainly of the lower extremities [1]. The overall prevalence of PAD is about 10% worldwide, and this increases to nearly 30% for patients older than 50 years. Patients with PAD may be symptomatic or asymptomatic. Up to 12% of older patients will have intermittent claudication which severely limits their activities in daily life [2]. The prevalence of asymptomatic PAD is much higher than symptomatic PAD, and several of the individuals with asymptomatic PAD will subsequently develop intermittent claudication leading to a predicted rise in the older population of Americans who will have ambulatory incapacity attributable to intermittent claudication [3].

Patients with PAD have an impairment in walking, exercise capability and quality-of-life. The treadmill graded exercise test is the standard procedure for evaluating functional loss and exercise capability in PAD patients [4,5,6]. However, this test is time consuming, expensive and not accessible in every medical facility. The evaluation of exercise capacity in congestive heart failure (CHF) and chronic obstructive pulmonary disease (COPD) is frequently done by a 6-minute walk test (6MWT) [1,7,8,9,10]. The severity of chronic pulmonary disease, cardiovascular disease, neuromuscular disease and advanced age is associated with a reduced 6-minute walk test distance [11,12,13]. The 6MWT is a reliable assessment of PAD with an intraclass correlation coefficient of 0.94 [14]. It has been shown that PAD patients with abnormal 6MWT have poorer clinical consequences, including higher vascular reactivity, risk of ischemic heart disease and limited physical activity [14].

PAD patients have sarcopenia with weakness and atrophy of skeletal muscle leading to a decrease in calf circumference, leading to functional loss in activities. We present a functional assessment of PAD involving the lower extremities. Ankle pressure is unreliable in the presence of medial arterial calcification present in PAD patients with concurrent diabetes and renal failure. Toe pressure measurements and pulse volume recordings (PVR) are not impacted to a significant extent. Therefore, amongst patients with noncompressible ankle vessels, toe–brachial index (TBI) is of significance. PAD limits patient ambulation and can be assessed by recording the distance walked on level ground in six minutes. 

The afore-mentioned variables were taken from a patient cohort and plotted against PAD severity in this study to determine the relationship, if any. However, there are multiple factors that can affect these measurements. These factors include—but are not limited to—CHF, COPD, diabetes, coronary artery disease (CAD), advanced age, arthritis and inflammatory joint diseases. All these co-morbidities have been found to be commonly associated with PAD due to similar risk factors. Therefore, in our study with so many confounding factors, we wanted to determine the relationship between a six-minute walk test and muscle circumference to severity of PAD.

A TBI less than 0.7 is considered abnormal. To our knowledge, there are no studies that determine any association between TBI and cardiovascular morbidity and mortality. There are no systematic analyses relating noninvasive assessment of PAD severity, symptoms of claudication and functional limitation (such as six-minute walk distance). We aim to assess relationships between severity of PAD, objectively measured effort tolerance using a six-minute walk distance and symptoms using a quality-of-life questionnaire. Clinically, this discordance between patient symptoms and severity of PAD is known; however, we assess these relationships using machine learning (ML) and use an ML ensemble to predict PAD. 

Statistical techniques of data analysis rely on stochastic data models. These tools are not as appropriate for multifactorial, complex relationships in large data sets. ML characterizes a powerful set of algorithms that can illustrate, modify, understand, predict and analyze large datasets, strengthening our understanding of PAD and our capability to predict with greater accuracy.

## 2. Methods

This paper includes prospectively collected data on 703 patients with established or suspected PAD, who underwent symptom assessments, toe–brachial indices’ and 6MWTs.

### 2.1. Patient Inclusion Criteria

All patients suspected or known to have PAD who presented to a busy, private practice were offered participation in a STRIDES program (described below). Written informed consent to participate in the program and permission for data analyses was obtained prior to conducting the STRIDES visit. The data collected was part of quality control.

### 2.2. Strides Protocol

A STRIDES room manned by a trained medical assistant was designated for the study protocol. Visits were conducted in the presence of a supervising physician. EKG (Electrocardiogram) monitoring of patients was not part of the protocol. Software designed by Newlifeware was an integral portion of the STRIDES program. Upon presenting for their STRIDES visit, patients were rested comfortably and followed a protocol wherein they would review and sign the informed consent. The test would be explained to them and they would answer the questions from the King questionnaire. Then, the medical assistant would remove the patient’s shoes and socks, ensuring that the feet were warm and toe pressure was recorded using plethysmography. Shortly thereafter, the calf circumference would be recorded and the arm pressure measured bilaterally. This was followed by measurement of 6MWT and reassessment of TBI after the test. Clinical photographs would be taken before and after completion of test.

The components of STRIDES included: informed consent; baseline photographs; modified King questionnaire (administered via Newlifeware software, www.newlifeware.com), which is a reliable questionnaire ready for use as an outcome measure in clinical trials [15]; bilateral measurement of maximum calf circumference as outlined by National Health and Nutrition Examination Survey Anthropometry (NHANES) Procedures Manual; measurement of baseline TBI using Hokansan MD35 mobile peripheral vascular system (essentially a photo plethysmograph); timed walk at a comfortable pace for the patient with no incline on a treadmill; measurement of post-6MWT TBI; patient education regarding PAD and significance of walking program, and a follow-up with a physician following completion of STRIDES visit. Data collected from STRIDES were stored in the Newlifeware database. This was provided in comma-separated format for statistical analysis. The data provided were stripped of all patient identifiers and deemed IRB-exempt by the West Virginia University IRB.

### 2.3. Data Analysis

Newlifeware data was available on a quarterly basis in comma-separated value format. RStudio Version 0.98.1062 software was used for analysis. Raw data had 832 entries and included data on training. Of the raw data, 709 had complete entries and comprise the data analysis presented herein. Standard data science techniques ensured data quality and consistency. Clean data was then transformed to long format [16], making analyses—grouped by various features—easy to accomplish. The raw data had 33 available variables. PAD (response variable) was evaluated using bilateral TBI prior to and following the 6MWT. Since there are no data available on post-6MWT TBI, this analysis addresses only the prestress baseline TBI. Other variables included response to the quality-of-life questionnaire categorized by question and, additionally, an aggregate score (average of scores for each response for the given patient) and calf circumference for each limb. The lower TBI recorded before the 6MWT and the lower calf circumference were included in the analysis. The aggregate score for the ten quality-of-life questions was scaled such that score 1 implied the fewest symptoms and 7 represented the worst symptoms. For the initial graphical analysis, data of all 703 patients were used. In the final analysis for ML, we chose to contrast patients with CLI (TBI < 0.4) against those with no PAD (TBI > 0.7) among 295 patients.

The supervised ML ensemble involved using PAD (TBI < 0.4) and no PAD (TBI > 0.7) as response variables and 6MWT (meters), calf circumference (cm), age (years) and total symptom score calculated from the questionnaire as predictor variables. Patients with TBI between 0.4 and 0.7 were not included in this analysis. Thus, this analysis would compare patients with severe PAD to those without PAD.

After cleaning the data, the features were run through a supervised ML ensemble comprising of Random Forest (RF), Neural Network (NN) and Generalized Linear Model (GLM) to determine which features would be best suited for the prediction of PAD using RStudio. The ensemble method was utilized as it would obtain better a prediction for PAD from the selected features compared to any of the afore-mentioned ML algorithms alone. An ML ensemble consists of a fixed set of alternative models; however, it allows for a flexible structure to exist among those alternatives. Essentially a layered ensemble was used whereby probabilities for PAD from the base layer were used as features predicting PAD in the higher layer. Sensitivities, specificities, positive predictive and negative predictive values, in addition to areas under the curve (AUCs), were obtained for all models of predictions to help gauge which set of features would be more predictive of PAD. Accuracy metric was not used to decide on accuracy of ML model.

RF is a classification method that is derived from a decision tree. A random sample is drawn from the data. Each of the variables/features are used at a node to split the data, with the positive responses going to the left of the tree. Using this classification method, the outcome-PAD in this case is predicted. Whether the tree is correct or not would be determined by a voting method. Those trees that are voted as being correct are kept. The method is re-iterated. Thus, a forest of decision trees develops; each of the trees may pick up on variability on one aspect of the data.

NN is highly structured and comes in layers. The first layer is the “input layer,” the final is the “output layer,” and all the layers in between are referred to as “hidden layers.” Thus, an NN can be depicted as a result of spinning classifiers coming together in a layered web. This is because each node in the hidden and output layers has its own classifier. The resulting sets of scores are then passed on as input to the next hidden layer for further activation and so on until it is reached to the output layer where the results of the classification are determined by the scores at each node to predict the likelihood of PAD. This happens for each set of inputs. This process is known as forward propagation or forward prop. Forward prop is an NN’s way of classifying a set of inputs.

GLM is a flexible generalization of linear regression which allows for response variables/features that have error distribution models other than a normal distribution. This model generalizes linear regression by allowing the linear model to be related to the response variable via a link function and by allowing the magnitude of the variance of each feature to be a function of its predicted value.

The area under a receiver operating curve (ROC)-AUC was used to assess each predictive model (RF with two different tuning parameters, NN, GLM and rpart). Highly correlated features were dropped. Features that did not increase the area under the ROC were dropped. The simplest ensemble that gave a reasonable ROC was retained. Care was taken not to overfit the model so as to keep the resultant model generalizable.

Mean and standard error of mean (SEM) for 6MWT distance was plotted against TBI, and a regression line was overlaid on the resultant scatterplot to graphically assess the relationship in the overall data. Data was categorized by severity of PAD using TBI and walking restriction, and the scatterplot was colored accordingly in order to visually assess the relationship amongst these variables.

## 3. Results

In the patients we studied, 19.6% (139/709) of patients had a TBI below 0.4 (critical limb ischemia (CLI)) and an average 6MWT distance of less than 200 m. We anticipated that all patients with severe PAD and TBI less than 0.4 would have reduced walk distances, but Figure 1 shows the correlation is not as linear as we anticipated.

One would anticipate decreased walking distances with increased perception of walking limitation. This patient perception of walking limitation is objectively borne out across the various categories of severity of PAD based on TBI and soft tissue loss (Figure 2). One would anticipate there would be more blue and purple colored dots in the patient panels with more severe PAD and within each panel we expect more of the blue and purple dots toward the right of the plot (assuming a more severe walking limitation would go with more severe claudication within each PAD category). Although such a relationship is suggested, it is not as clear. Visually there appear to be more blue and purple dots in the patients with moderate PAD and not as much in those categorized as CLI. This suggests complicated relationship(s) between 6MWT, perception of walking limitation and symptom severity across different PAD severities, which may be nonlinear.

Summarizing the above two approaches of noninvasive categorization of severity of PAD using TBI and walk distance or TBI and tissue loss, the three features (TBI, walking limitation and claudication) do have a relationship. There are numerous interactions amongst these features that may or may not be linear. In order to explore the nonlinear relationships, ML techniques were brought into play.

To assess the relationship between PAD symptoms and functional limitation, total symptom score was plotted against 6MWT distance (Figure 3). Total symptom score is a sum of all the symptoms collected using the data collection tool in the STRIDES program. Individual question scores were added and divided by the number of questions answered to come up with a composite score for each patient. Amongst patients without PAD as symptoms worsen (reflected by an increase in the score), there is a slight reduction in the 6MWT distance (Figure 3). Amongst patients with PAD, those with the least symptoms have the most functional limitation. Stated another way, those who do not walk much have fewer symptoms, those with moderate symptoms are able to walk more than 200 m. Those with more severe symptoms have a walking limitation (less than 200 m). Thus, severity of symptoms has a nonlinear relationship with walking limitation, and this relationship appears to change direction amongst PAD patients.

Table 1 represents various measures of the 6MWT distance, less than 200 m being considered test-positive to assess PAD stratified across various symptom-score categories (Figure 3). For symptom scores less than 6, the walk test maintains a high specificity of greater than 91% and positive predictive values of 66%.

If we were to use a mean and standard error (SE)-bar plot to assess the relationship between PAD status and predictor variables (Figure 4), no apparent relationship would be evident. If we had relied on statistical methods to assess these complex relationships, we would have concluded absence of relationship(s) between the severity of PAD and total symptom score, calf circumference, 6MWT distance or age in decades. Hence, this is based on univariate analysis, precluding the need for further data analysis.

## 4. Discussion

### 4.1. Machine Learning

#### 4.1.1. ML Versus Statistics

Table 2 presents a quick comparison of statistics and ML. In statistics we assume a structure to the data. Most models take response and predictor variables as being related to each other. Parameters for this relationship are developed using mathematical techniques. The predicted relationship identified is assumed to have randomly distributed noise superadded to it. In the ML realm there are no assumptions made of the relationship between response and predictor variables.

Once a mathematical model is developed, statistics uses a goodness of fit measure or examines the residuals to decide which model to pick. In ML, various measures of predictive accuracy are used to decide model selection. In this instance, we use the area under a receiver operating curve as the measure.

Statistics assumes that a mathematical model emulates data in nature. In ML, once we use an ensemble of models and pass on probabilities from one layer to the next one as features, the model becomes a ‘black box’. No mechanistic ‘understanding’ of various relationships can be achieved. As we illustrate in this case, we develop a relationship using ML techniques to tease apart nonlinear relationships. These relationships change directionality and interact; that is to say, patients with more severe PAD ambulate less and thereby fail to get claudication.

#### 4.1.2. ML Ensemble

ML identifies patterns in existing data using statistical learning and computer techniques. The resultant model can then be used to predict the outcome of interest. This is initially done in a portion of the data that has been ‘held out’ and, subsequently, been used on unrelated data wherein the outcome is unknown to predict outcomes. The use of multiple algorithms to obtain a better predictive performance is termed an ensemble. Data is split into testing and training datasets. The outcome (PAD) is known in both these sets. The RF ML model draws a sample from the training dataset and uses a classification scheme to predict PAD. A number of decision trees are formed. The better performing trees are voted out of the RF and a final model is developed. Using different settings, two separate RF models were used. An NN algorithm is inspired by the structure and functional aspects of biological NN. Such a model had a slightly inferior AUC in comparison to the RF model. A GLM fits a linear model to the data as done in traditional statistics. Recursive partitioning considers every possible feature in the data set and the partitions generated using it.

Each of these techniques has its own theoretical limitations. However, using them together could result in a better predictive model and a greater AUC. The model then gives relative importance to each of the features used in the model on a scale. Table 3 represents AUCs for each of the ML models and the ensemble. The ensemble with its individual components and their relative weights is also shown. Note though in this instance, the AUC of random forest 1 (0.69) and ensemble (0.687) is almost the same. Also, the ensemble performed similarly in the training and validation datasets. 

### 4.2. Clinical Significance

Using detailed history, calf circumference measurements, 6MWT and patient’s age, one could predict patients with CLI. In North America, 500–1000 CLI patients per million are expected. Of the patients with CLI, 30% endure a major amputation, 25% will not survive and 20% suffer with pain or tissue loss within the first year [18,19]. A year later, CLI is expected to have been resolved in only a quarter of the patients. Therefore, early recognition and treatment of CLI is pertinent. Our findings of using simple clinical measures to identify patients at risk of CLI would allow early referrals to a more resource-intensive test and a consultation with the specialist. Treatment modality would include risk-factor modification, smoking cessation, exercise program and endovascular intervention if deemed necessary. Based on Table 1, for lower symptom sores (<6), using the 6MWT has a high specificity of 91% and a positive predictive value (PPV) of 66%, which signifies that a positive 6MWT for patients with low symptom scores could be suggestive of PAD leading to high likelihood of confirming disease. This obviates the need for both angiogram testing and TBI measurements in asymptomatic patients. Our results, therefore, prove to be a useful tool for diagnosing PAD in asymptomatic patients considering the ease with which both the modified questionnaire and the 6MWT can be administered. We do not advocate routine angiography in these patients, but we suggest this as a test to find patients at two to fifteen-fold increased risk for cardiovascular events with significant incidence of limb loss, as seen in CLI patients. In comparison to Framingham risk score or stress testing, this is simpler as no blood work is needed, and it uncovers a high-risk population. 

Based on our ML model, symptom score appears to be the most significant variable (Figure 5) accounting for the variability in the model predicting PAD. This seems consistent with the clinical realm where intervention is performed for patients that have lifestyle-limiting claudication [20]. The next most important variable in our model was the calf circumference. This is a measure of macroscopic tissue loss. Per guidelines, in patients with CLI, revascularization is to be performed to minimize tissue loss (implying nonhealing ulcers). Our data however suggests macroscopic tissue loss has an independent significance in PAD prediction with a greater tissue loss implying severe PAD. Whether the tissue loss is related to reduced blood flow or reduced activity or both is unclear. Whether intervention of PAD would improve tissue loss is an interesting unknown. Age, as a significant variable in our model, is a finding consistent with previous literature [21].

Our model also identified 6MWT distance as a predicting variable albeit of least importance (Figure 5). A pertinent question, however, is that these patients often have multiple comorbidities (CAD, COPD or chronotropic incompetence). The lack of a strong relationship between 6MWT distance and PAD severity implies that the comorbidities may interact with 6MWT distance, exerting effect modification and that intervening on PAD, which is not limb-threatening, may not have an impact on overall patient outcome. Conversely, if our model had demonstrated a strong relationship between 6MWT distance and PAD severity, one could have argued strongly about intervention on severe PAD patients that is not limb threatening. Our data, however, suggest that since the relationship between 6MWT distance and PAD severity is not as significant, intervention to improve quality of life may not be desirable. However, further research with measurement of baseline characteristics and postintervention assessment would help address this question better.

Although our ML ensemble model has a modest predictive accuracy (AUC 0.69) at best in predicting PAD, it is pertinent to note that it allows a quantitative assessment using predictive variables including symptom score, calf circumference, age and 6MWT (Figure 5). This accommodates the nonlinear relationships between PAD severity, functional limitation and symptom score which is often seen clinically in the form of discordance between patient symptoms and their disease severity. As shown in Table 1, in patients with low symptom score (<6), 6MWT could be used as a screening tool for PAD (specificity 92% and sensitivity 26%). Combining this with ASCVD score, one may be able to identify the patient population most likely to be at risk for cardiovascular adverse events. Based on our results, a 6MWT could be used to identify PAD without performing ABI which is not always readily available in a primary care physician’s office.

## 5. Conclusions

Our data analysis illustrates how ML allows exploration of nonlinear relationships. ML showed that symptom severity assessed by the quality-of-life questionnaire is a variable of high importance amongst patients with PAD. 6MWT distance is also an important variable, however confounding factors including diseases not considered in our analysis likely impact it. The resultant model using a simple questionnaire, calf circumference, age and 6MWT distance predicts critical limb ischemia with an AUC of 0.69. For low symptom scores, we also found high specificity of 6MWT for PAD, which is suggestive of its use as a screening test.

## Figures and Tables

**Figure 1 diagnostics-10-00515-f001:**
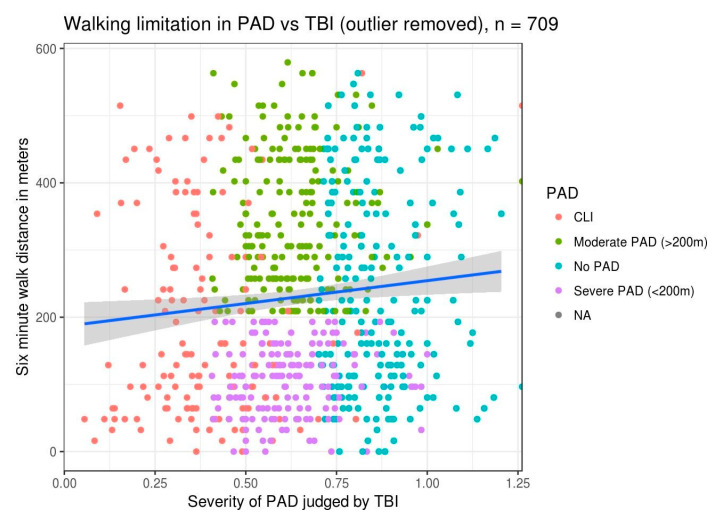
Six-minute-walk-test (6MWT) distance versus toe–brachial index (TBI): This illustrates a slight improvement in walk distance with improvement in TBI values. The scatterplot of individual patient data is colored by severity of peripheral arterial disease (PAD), as outlined in the legend. A linear regression line illustrating the relationship between walk distance in meters and TBI is overlaid on the data. CLI: critical limb ischemia.

**Figure 2 diagnostics-10-00515-f002:**
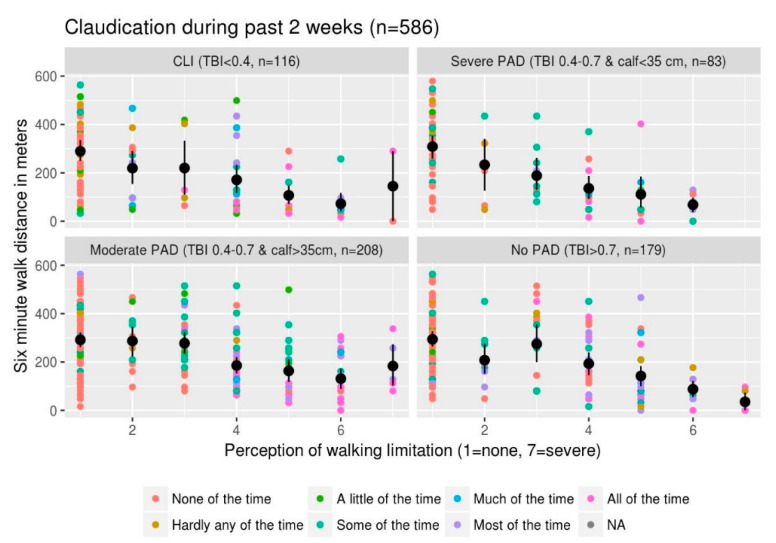
6MWT distance versus perception of walking limitation: This illustrates the relationship between 6MWT distance, perception of walking limitation from quality-of-life questionnaire and symptoms of claudication in the last 2 weeks. The scatterplot of six-minute walk distance versus perception of limitation where each dot represents a single patient is colored by severity of symptoms using the color scale specified in the legend. The black dots are the mean 6MWT distance for each given category of perception of walking limitation with the bars representing the standard error of mean (SEM). Each panel represents data from patients with various severities of PAD ranging from CLI in the top left panel to no PAD in the bottom right panel. We used a combination of severity of TBI and calf circumference to categorize PAD severity. Calf circumference of 35 cm is based on an article by Ali et al. [17]. The colored dots represent data from individual patients with the color code exhibiting the severity of claudication in the past 2 weeks with the blue/purple colors representing more severe symptoms.

**Figure 3 diagnostics-10-00515-f003:**
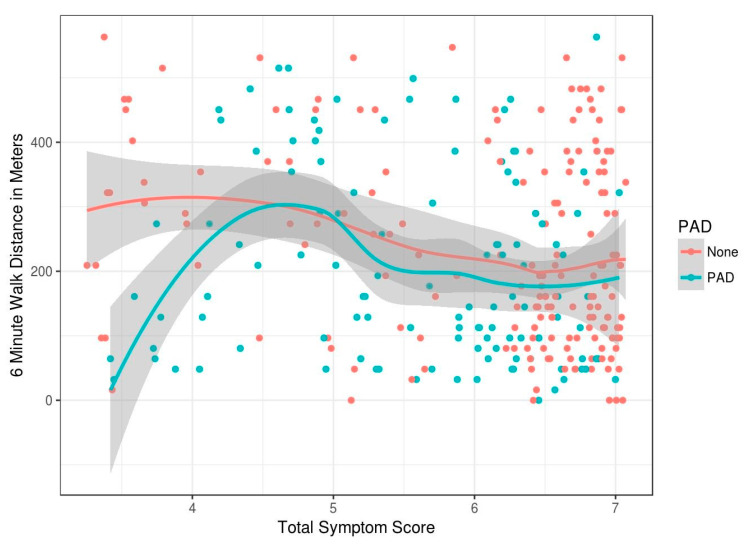
Shows the mean and standard error (SE) bars for the continuous features and illustrates this.

**Figure 4 diagnostics-10-00515-f004:**
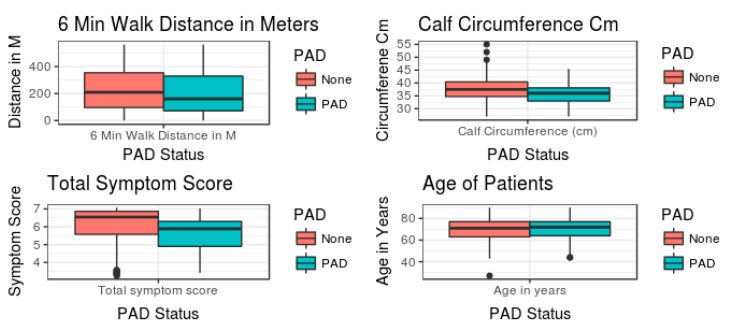
Mean ± SE of the predictor variables between no PAD and PAD.

**Figure 5 diagnostics-10-00515-f005:**
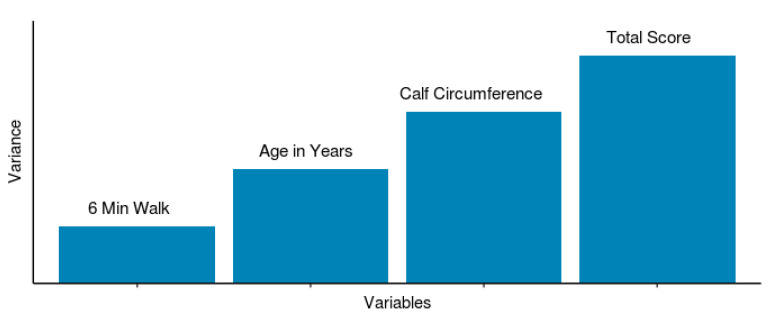
Predictive variables: final ML model predictor variables with relative variance contribution to PAD.

**Table 1 diagnostics-10-00515-t001:** Specificity of diagnosing critical limb ischemia using six-minute walk distance and symptoms score alone.

	Symptom Score (x) Categories with 6MWT			
	<4	<5	<6	6–7
Sensitivity	6.89%	11.3%	26.0%	28.4%
Specificity	90.5%	96.6%	91.6%	63.6%
Positive Predictive Value (PPV)	32.0%	68.4%	66.6%	33.6%
Negative Predictive Value (NPV)	60.0%	62.9%	65.8%	57.8%

**Table 2 diagnostics-10-00515-t002:** Contrasting statistics and machine learning techniques.

	Statistics (Data Modeling)	Machine Learning (Algorithmic)
Data Structure	Response variables = function (predictor variables, random noise, parameters)	Exact relationship between response variables and predictor variables is unknown
Model Validation	Uses goodness of fit and/or examines the residuals	Use predictive accuracy
Limitations	Mathematical model emulates nature’s model	A good predictive model may use features derived from variables thus a ‘black box’

**Table 3 diagnostics-10-00515-t003:** Area under the receiver operator curve for models predicting critical limb ischemia.

Machine Learning Model	ROC AUCs
Random forest 1 (rf1)	0.69
Random forest 2 (rf2)	0.63
Neural network (nn)	0.63
Generalized linear model (glm)	0.68
Recursive partitioning (rpart)	0.64
Ensemble = rf1 (1.30), rf2 (−1.70), nn (0.09), glm (−0.13), rpart (−0.70)	0.687~0.69

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
