# Peer review of "Machine Learning Confirms Nonlinear Relationship between Severity of Peripheral Arterial Disease, Functional Limitation and Symptom Severity"

_diagnostics, 2020, doi:10.3390/diagnostics10080515_

Round 1
Reviewer 1 Report
This paper is ready to publish.
Author Response
Please see the attached full manuscript.

Reviewer 2 Report
The authors have addressed the previous comments.
I suggest the Authors to include paragraph /section in the introduction and discussion regarding how this work will be clinically relevant. It is quite well recognised patients often display discordance in their symptoms as compared to their arterial disease severity (ie, patients with significant arterial occlusion may be relatively symptom free, whereas short segment arterial occlusion can cause severe symptoms in some). This is one of the key message of this paper - which in itself is not novel. How does the study shed new light to this recognised clinical manifestation of disease?
Author Response
Please see the attachment.

This manuscript is a resubmission of an earlier submission. The following is a list of the peer review reports and author responses from that submission.
Round 1
Reviewer 1 Report
The authors do a good job in establishing the rationale/aims for this study as well as its clinical relevance. They documented the weak and at times non-linear relationships between potentially relevant variables, which highlights the benefit of attempting a machine learning method. They also explain the difference between a classical statistical and a machine learning approach and why the latter is better suited for this problem.
An area that needs further explanation is the results section for the Machine Learning Ensemble method. This information found in this section doesn’t match with data found elsewhere in the manuscript. In the conclusion, the authors highlight that the ROC AUC for this model is 0.66, which is worse than the Random forest 1 and Generalized Linear Model. If the individual methods perform better to predict PAD, is the ensemble method needed? Since there is not much information on the performance of this combined model, it makes this section hard to understand. Another point of confusion is how the relative importance of the weights were obtained. I know, this information originates from the output of the model but it needs to be clarified. The strength of this ensemble approach would be more apparent if the relative weights for each of the constituent models would also be included. Finally, it was mentioned earlier that the data was split into training and validation sets. It would be important to include how this model performs on the validation set of data in order to comment on its generalizability.
Author Response
We thank the reviewer for the detailed comments which certainly help make the paper better.
Comment: ‘An area that needs further explanation is the results section for the Machine Learning Ensemble method. This information found in this section doesn’t match with data found elsewhere in the manuscript. In the conclusion, the authors highlight that the ROC AUC for this model is 0.66, which is worse than the Random forest 1 and Generalized Linear Model. If the individual methods perform better to predict PAD, is the ensemble method needed? Since there is not much information on the performance of this combined model, it makes this section hard to understand. Another point of confusion is how the relative importance of the weights were obtained.’
Response: We thank the reviewer for the comment. The ensemble consisted of the following individual ML algorithms: rf1, rf2, nn, glm and rpart. And this was highlighted in the discussions section as required by the reviewer. While we highlighted the significance of using ML ensemble rather than individual ML algorithms for prediction for greater accuracy, it is noteworthy to state that the AUCs for all the individual algorithms and the ML ensemble are almost the same (Table 3) ~ 0.6 with the AUC for Rf1 and Ensemble being 0.69 and 0.687 respectively. As per the reviewer’s comment, we also included the relative weights of the individual components of the ensemble (Table 3). We employed the ensemble but, in this instance, rf1 could have been selected as well given the similar predictive accuracy. Interesting to note, that rf1 is weighted the maximum as an ensemble constituent as shown in Table 3. The changes are reflected as highlighted in the manuscript in the discussions and conclusion section.
Comment: ‘Finally, it was mentioned earlier that the data was split into training and validation sets. It would be important to include how this model performs on the validation set of data in order to comment on its generalizability.’
Response: We thank the reviewer for the comment. The ML models performed similarly in the validation datasets. And therefore, we commented on the generalizability of the model in the discussions sections to highlight this.
Reviewer 2 Report
This wide observational study concerns a population of 703 patients with established or suspected peripheral arterial disease (PAD). Aim of the observation is to explore the relationship between self-evaluation of a symptomatic score (through a disease-specific validated quality of life questionnaire, the modified King questionnaire), and some of the main objective clinical diagnostic parameters or markers of PAD, as the toe-brachial index (TBI) the bilateral maximal calf circumference of legs (CCH) and the measurement of the walking distance (6 min WD). Researchers mainly used machine-learning procedures, maintaining that results so obtained are independent from rational and "mechanistic" prejudice. In fact, "in the machine-learning realm there are no previous assumptions about the relationships, or mechanistic understanding of the various relationships found". Results indicate that this type of data analysis allows demonstration of non-linear relationships between the different data, whereas traditional statistic evaluation would have found no-correlation. As an example, symptoms score severity (by questionnaire) is found to be by far the most important diagnostic bio-marker, followed by calf circumference, age, and 6 min walking distance. However, severity of the symptom score shows a non-linear relationship with the measurement of walking distance: in fact, the walking distance test has higher specificity and positive predictive value in subjects with lower symptom score or asymptomatic at rest. Other strong or loose correlations are documented in the paper.
Comment. The low correlation between patient referred symptoms and the measurement of walking distance, as well as other variables having low or no intercorrelations in PAD, is a concept well known by the clinicians. Lack of correlation between self-reported symptoms and objective bio-markers is a concept certainly not limited to PAD! Many different factors as co-morbidities, age, request of physical performance, habitual exercise, type and location of the atherothrombotic lesions and many other factors can interfere. I quote as extreme paradox the fact that in critical limb ischemia walking limitation is irrelevant and in acute ischemia the calf circumference is also irrelevant. Thus the results cannot be defined absolutely "original". However, it is important to have a mathematical confirmation of well known clinical concepts, based on a new type of statistical methodology. Another positive contribution of this paper is the stress given of the calf circumference as a neglected biomarker of PAD indicating muscular tissue loss.
Author Response
Attachment.

Reviewer 3 Report
The authors used Machine Learning to analyze the relationship between the severity of the peripheral arterial disease, functional limitation and symptom severity and found that the nonlinear relationship between them. The comments are provided to improve the manuscript.
Major comments:
- Could you please explain why you exclude 123 patients in the analysis of Figure 2 because the number of analysis is 586? It is also not clear the purpose (why you perform this analysis) and conclusion for Figure 2?
- It is not clear that the advantage of Machine learning compared to statistical analysis. If you used statistical techniques of data analysis for the same set of data, what is the conclusion? Is it the same as when you used Machine learning?
- It is better to have subheading in the main text. Each figure and table should have a title followed by a short explanation.
Minor comments:
- Please define CAD when you first used it in the Introduction section.
- What the number stands for in y-axis of Figure 4?
- How do you define the importance in Figure 5?
Reviewer 4 Report
It is an interesting paper using algorithmic ML to show that symptom score, calf circumference, age in years and 6MWT are the variables of importance for PAD. It is of significance to explore the non-linear relationships among different variables to predict PAD.
There are couple concerns for the authors' considerations:
(1) The authors have a main focus on 6MWT; However, 6MWT shows the least significant and poor sensitivity for PAD prediction. Is there any evidence showing PAD patients may attempt to walk or practice more intentionally to overcome the potential symptoms? The authors may consider to explain more regarding the reasons accounting for this data;
(2) The author shall put more analysis in detail to explain how score, muscle circumference are more related to PAD severity as shown in Figure 5.
(3) Full names for TBI, ML and 6MWT in the instruction;
(4) There is no figure legend for Figure 4;
(5) Figure 5 should be put into the result section; "Figure 3-Top 10..." should be deleted from the Figure 5.
(6) The data analysis section in Method is very lengthy. A detailed explanation of RF, NN and GLM was performed; however, very brief data were shown in the contents.
Round 2
Reviewer 2 Report
Authors accepted my comments and made corresponding changes in the text, including a change in the title. I think therefore, that the paper is of interest and now deserves publications.